# Shotgun Metagenomic Sequencing to Assess Cyanobacterial Community Composition following Coagulation of Cyanobacterial Blooms

**DOI:** 10.3390/toxins14100688

**Published:** 2022-10-07

**Authors:** Kim Thien Nguyen Le, Juan Francisco Guerra Maldonado, Eyerusalem Goitom, Hana Trigui, Yves Terrat, Thanh-Luan Nguyen, Barry Husk, B. Jesse Shapiro, Sébastien Sauvé, Michèle Prévost, Sarah Dorner

**Affiliations:** 1Department of Civil, Geological and Mining Engineering, Polytechnique de Montréal, Montréal, QC H3C 3A7, Canada; 2Department of Geography and Environmental Studies, Toronto Metropolitan University, Toronto, ON M5B 2K3, Canada; 3Institut National de Santé Publique de Quéquec, Montréal, QC H2P 1E2, Canada; 4Department of Biological Sciences, University of Montréal, Montréal, QC H2V 0B3, Canada; 5BlueLeaf Inc., 310 Chapleau Street, Drummondville, QC J2B 5E9, Canada; 6McGill Genome Centre, McGill University, Montréal, QC H3A 0G1, Canada; 7Department of Microbiology and Immunology, McGill University, Montréal, QC H3A 2B4, Canada; 8Department of Chemistry, University of Montréal, Montréal, QC H3C 3J7, Canada

**Keywords:** cyanobacterial blooms, cyanobacterial community, coagulation, cyanotoxins, ferric sulfate, high-throughput sequencing, mesocosms, microcystins, shotgun metagenomics

## Abstract

The excessive proliferation of cyanobacteria in surface waters is a widespread problem worldwide, leading to the contamination of drinking water sources. Short- and long-term solutions for managing cyanobacterial blooms are needed for drinking water supplies. The goal of this research was to investigate the cyanobacteria community composition using shotgun metagenomics in a short term, in situ mesocosm experiment of two lakes following their coagulation with ferric sulfate (Fe_2_(SO_4_)_3_) as an option for source water treatment. Among the nutrient paramenters, dissolved nitrogen was related to *Microcystis* in both Missisquoi Bay and Petit Lac St. François, while the presence of *Synechococcus* was related to total nitrogen, dissolved nitrogen, dissolved organic carbon, and dissolved phosphorus. Results from the shotgun metagenomic sequencing showed that *Dolichospermum* and *Microcystis* were the dominant genera in all of the mesocosms in the beginning of the sampling period in Missisquoi Bay and Petit Lac St. François, respectively. Potentially toxigenic genera such as *Microcystis* were correlated with intracellular microcystin concentrations. A principal component analysis showed that there was a change of the cyanobacterial composition at the genus level in the mesocosms after two days, which varied across the studied sites and sampling time. The cyanobacterial community richness and diversity did not change significantly after its coagulation by Fe_2_(SO_4_)_3_ in all of the mesocosms at either site. The use of Fe_2_(SO_4_)_3_ for an onsite source water treatment should consider its impact on cyanobacterial community structure and the reduction of toxin concentrations.

## 1. Introduction

Cyanobacteria are among the most ancient organisms on Earth, and they are present in aquatic environments worldwide [1,2]. When intense blooms appear, cyanobacteria can produce large quantities of toxins. The increasing frequency and intensity of the toxic cyanobacterial blooms have affected wild animals, domestic animals, aquatic ecosystems, and human health [1,3]. Microcystins (MC), nodularin (NOD), cylindrospermopsin (CYN), and anatoxin-*a* (ANTX-*a*) are frequently found in freshwater blooms, while microcystins are the most well-known and reported toxins [1,4].

Given the harmful nature of cyanobacterial blooms for recreational and drinking water uses, various types of treatments have been proposed, including their coagulation, oxidation, and adsorption with powdered activated carbon [3,5,6]. The effectiveness of coagulation on cyanobacterial cells and cyanotoxin removal has been studied [3,7,8], and it has been demonstrated to be effective for the removal of cyanobacterial cells and cell-bound toxins [3]. However, cyanobacterial cells may be lysed, and their toxins released into the supernatant during the drinking water treatment sludge storage processes [9]. Several studies have applied high-throughput sequencing to investigate the microbial diversity and community structure during coagulation [10,11]. For example, a recent study applied high-throughput sequencing to microbial communities in sludge from six different drinking water treatment plants, and the authors of this found that there was a relationship between the toxic cyanobacteria and the pathogens in the sludge resulting from raw water with nutrients and iron (Fe) [11]. Similarly, Pei et al. 2017 [10] studied the microbial structure in drinking water treatment sludge that was formed by different types of coagulants, including AlCl_3_, FeCl_3_, and polyaluminium ferric chloride (PAFC). Their results indicated that after four days of storage, the relative abundance of the dominant genera *Microcystis*, *Rhodobacter*, and *Phenylobacterium* decreased, thus leading to an increase in the quantity of the extracellular microcystin and organic matter [10].

Shotgun metagenomic sequencing has been used to understand the cyanobacterial diversity in sludge and the risk of toxin release after conventational drinking water treatment processes [8]. This study showed that there was an accumulation of cyanobacterial cells, as well as potential cell growth, in the sludge from the raw water influx as a result of coagulation and sedimentation in the drinking water treatment process. If the cyanobacterial communites in the sludge supernatant are then recycled, the cyanobacteria and cyanotoxins could thereby be introduced to the intake water. The sludge cyanobacterial communites were positively correlated with the total nitrogen (TN), total phosphorus (TP), particulate phosphorus (PP), and total organic carbon (TOC) [8].

Understanding the cyanobacterial composition structure before and after coagulation occurs will offer an insight into the mechanisms of coagulation and provide a stronger basis for predicting treatment efficacy. Most of the studies on this subject have been conducted at the laboratory scale; however, the cyanobacterial communities might be dynamically affected by various environmental conditions. With the exception of the study that was conducted by Le et al. 2021 [12] where the cyanobacterial fate after the coagulation in the mesocosms was assessed using taxonomic cell counts, there have been no studies that have been conducted on the exploration of the cyanobacterial communities in water after the coagulation in fresh blooms in the source water. Such information can be used to guide source water treatment options. In order to improve the understanding of the cyanobacterial communities subsequent to the source water coagulation, this study aims to: 1) investigate the composition structure of the cyanobacteria using shotgun metagenomics sequencing in, in situ mesocosms before and after the coagulation of the cyanobacterial blooms over a short time period; 2) compare the shotgun metagenomics sequencing results with the microscopic taxonomic cell counts; 3) determine the relationship between the environmental parameters and the cyanobacterial communities that are associated with cyanotoxins before and after their coagulation with ferric sulfate (Fe_2_(SO_4_)_3_).

## 2. Results and Discussions

### 2.1. Cyanobacterial Community Composition Assessed by Shotgun Metagenomic Sequencing

The cyanobacterial communities in the mesocosms were studied at the beginning (T0) and after two days (T48) of the study period by using shotgun metagenomic sequencing at the genus level in Petit Lac St. François (PLSF) in 2019 and Missisquoi Bay (MB) in 2018 and 2019. The number of reads were normalized by their relative abundance (Figure 1). The metagenomic results were validated for the experiments on 08–10 August 2018 and 13–15 August 2019 in MB; on 24–26 July 2019 and 05–17 August 2019 in PLSF.

In MB, *Dolichospermum* was the dominant genus (approx. 25–30% of the total relative abundance) and *Mi**c**rocystis*, *Nostoc*, *Fischerella*, and *Calothrix* were the representative genera in all of the mesocosms at T0 on 8 August 2018 and 13 August 2019 (Figure 1A,B, respectively). These results are in accordance with those of several previous studies which indicated that *Dolichospermum* was a cyanobacterial genus that was frequently found in the blooms in MB [3,6,13,14]. After two days, on 10 August 2018, there was no change in the relative abundance in any of the mesocosms in comparison with the initial composition (Figure 1A). This observation corresponds with that of a previous report, thereby demonstrating the stability of the cyanobacterial community composition at the genus level after two days of oxidation with H_2_O_2_ (20 mg/L) and CuSO_4_ (2 mg/L) in the control mesocosms and the mesocosms with an oxidant in Petit Lac St. François [15].

However, a remarkable change in the cyanobacterial composition at the genus level was observed in both the control and the coagulated mesocosms at T48 on 15 August 2019 in MB (Figure 1B). The most abundant genus shifted from being *Dolichospermum* to being *Synechococcus* (approx. 45% of the total relative abundance) and *Microcystis* (38.7% of the total relative abundance) in the control and the coagulant mesocosms, respectively. This shift can probably be attributed to the marked decrease of the *Dolichospermum* cell count (greater than 97% reduction) in all of the mesocosms (Appendix A). The loss of *Dolichospermum* biomass in the control mesocosms from 13 August to 15 August in 2019 may be the result of a drop in pH from 8 to 6 which occurred (Appendix A), thereby potentially interfering with the cyanobacterial growth, in general [16]. This lower pH level could contribute to the dominance of *Synechococcus* in the control mesocosms because it can flourish at pH 6.5 [17,18]. However, we did not observe such an abundance of *Synechococcus* in the mesocosms at the doses of 20 mgFe/L and 35 mgFe/L, thereby implying that the *Synechococcus* cells were well captured by the coagulation process. *M**icrocystis* became the dominant genus, although both *Dolichospermum* and *Microcystis* were largely removed (>86%) in the coagulated mesocosms on 15 August 2019 (Appendix A). This suggests that cell survival and re-growth might have occurred during the two days of the sludge storage process [8]. The competition between the *Microcystis* and *Dolichospermum* strains at the laboratory scale was investigated, and it was found that *Microcystis* significantly inhibited the growth of *Dolichospermum* [6].

In PLSF, at T0, *Microcystis* was the predominant genus, accounting for 30–37% of total relative abundance that was documented on 24 July 2019, while the highest cyanobacterial relative abundances consisted of *Dolichospermum* (9–10.5%), *Synechococcus* (8–12%), and *Microcystis* (10–12.5%) on 5 August 2019 in all of the mesocosms (Figure 1C,D). The relative abundance of the dominant genera remained stable in the control mesocosms at T48 on 26 July and 7 August in 2019 in PLSF. In the mesocosms that had the addition of 20 mgFe/L and 35 mgFe/L, the relative abundance of *Microcystis* decreased to 11.5% and 10%, respectively, at T48 on 26 July 2019 (Figure 1C). This is coherent with the trends that were observed in the taxonomic cell count results on 26 July 2019 in our previous study [12]. On 07 August 2019, the reduction of 100% of the *Dolichospermum* in the taxonomic results (Appendix A) may be the result of a notable increase in the relative abundance of *Microcystis* in the mesocosms that had 35 mgFe/L added to them (Figure 1D).

The results of the microscopic taxonomic cell counts and shotgun metagenomics sequencing do not completely correspond with each other in this study (Figure 1, Appendix A and Appendix A). *Dolichospermum* and *Microcystis* were detected by both of the approaches, but *Aphanocapsa*, *Aphanothece*, *Coelosphaerium*, *Merismopedia**,* and *Chroococcus* were detected only by microscopic taxonomic cell counts. Moreover, *Aphanizomenon* was identified to be predominant by the taxonomic cell counts, while the metagenomics test found that *Dolichospermum* was the most dominant genus at the beginning of the sampling period in all of the mesocosms on 13 August 2019 (Figure 1B and Appendix A). These observations correspond with previously reported findings wherein the microscopic taxonomic cell counts and high-throughput sequencing do not completely match [6,8].

The limitations of microscopic and shotgun metagenomics approaches may provide an explanation for the differences in the structure of community composition that were found between these methods. Small-sized and morphologically similar species may escape their detection by microscopy [19,20]. The significant morphological deformation of cyanobacteria species by the coagulation or the oxidation process may also hinder the ability to identify these cells [6,8,21]. Shotgun metagenomics (high-throughput sequencing) is a powerful tool for microbial species or community detection and classification to observe the microbial dynamics through water treatment processes [8]. A wide variety of taxonomic and functional classifier pipelines have been developed and are available to make the analysis of metagenomic sequencing more accessible than they ever have been before. However, the accurate performance of these analyses may be hindered by the process of DNA extraction [20,22], frequent taxonomy database changes over time, and the number of false positive taxa predictions [23].

### 2.2. Impact of Coagulation on Cyanobacterial Richness and Diversity

#### 2.2.1. Principal Component Analysis on the Relative Abundance of Cyanobacterial Community

To gain an insight into the changes in the cyanobacterial community composition dynamics before and after the coagulation in MB and PLSF, we performed a principal component analysis (PCA), the results of which are shown in Figure 2 and Appendix A.

In MB, PC1 and PC2 explained 89.4% and 6.12%, respectively, of the variation in the cyanobacterial community composition among the samples (Figure 2). The cyanobacterial composition at the start of the study period (T0) on 8 August 2018 and 13 August 2019 was clustered in the blue circle and the orange circle, respectively (Figure 2A), and comprised of mainly *Dolichospermum*, as shown in Figure 2B, which supports the result that are shown in the bar graph (Figure 1A,B). Our result is in accordance with that of a previous paper which reported that the community composition did not vary considerably from year-to-year in Missisquoi Bay [14]. After 48 h, on 10 August 2018 and 15 August 2019, there were two different trends in the cyanobacterial composition in the mesocosms. On the one hand, in all of the mesocosms, the cyanobacterial composition clustered in the same blue circle, which is similar to the initial conditions that were observed on 10 August 2018 (Figure 2A). On the other hand, the cyanobacterial composition displayed a progressive shift in both the control and coagulated mesocosms on 15 August 2019. *Synechococcus* became more abundant in the control mesocosms (Figure 2A,B), which is in agreement with the results that are shown in the bar graph (Figure 1B). For the mesocosms with doses of 20 and 35 mgFe/L, the cyanobacterial community composition shifted mainly from being dominated by *Dolichospermum* to being dominated by *Microcystis* (Figure 2A,B). This result is in agreement with the recently reported results of the changes in the community structure at the genus level that were monitored in the samples that were retrieved from raw water and sludge supernatant after a coagulation process in a drinking water treatment plant [8].

Appendix A presents the PCA analysis of the cyanobacterial community composition following its coagulation using different doses which we performed in PLSF. PC1 and PC2 explain 54.0% and 41.6% of the variation in the cyanobacterial community composition among the samples, respectively. The cyanobacterial composition at T0 on 24 July 2019 and 5 August 2019 were clustered separately into two groups (blue and orange, respectively), thus implying that the cyanobacterial communities were distinct (Appendix A). At T0 of the study period, on 24 July 2019, in all of the mesocosms, *Microcystis* was the predominant genus, while on 5 August 2019, the cyanobacterial composition comprised mostly of *Synechococcus*, *Dolichospermum*, and *Microcystis* (Appendix A). After 48 h, on 26 July 2019, the cyanobacterial community composition was similar to that of the initial community composition in the control mesocosms, whereas it experienced a substantial shift from *Microcystis* to *Synechococcus* and *Dolichospermum* in the mesocosms containing doses of 20 and 35 mgFe/L (Appendix A), thereby confirming the results that are shown in the bar graph (Figure 1C,D). Similarly, after 48 h in the control mesocosms on 5 August 2019, the cyanobacterial composition was similar to that of the inital composition. A similar trend was observed in the mesocosms that had a dose of 20 mgFe/L, while the cyanobacterial composition changed in the mesocosms containing the dosage of 35 mgFe/L.

The PCA analysis was also utilized to evaluate the relationship between the cyanobacterial community and the cyanotoxins (Appendix A). A relationship between the total intracellular microcystin (ΣMCs) and the intracellular toxin MC-LR with the genus *Microcystis* which was detected through metagenomics was found at T0 and T48 in all of the mesocosms. This is consistent with our previous observation using taxonomic cell counts where we found that the highest ΣMCs occured in the periods when *Microcystis* was dominant [12]. It also suggests that the potentially toxigenic genera that were correlated with the microcystin concentrations were *Microcystis* and *Dolichospermum* in a study of 22 lakes in southern Quebec [24]. In addition, a recent study observed that Fe_2_(SO_4_)_3_ can remove more than 93% of the *Microcystis* cells with a dose of 35 mgFe/L in a mesocosm experiment [12]. Therefore, the application of Fe_2_(SO_4_)_3_ may be considered to be an efficient onsite barrier against toxin-producing cyanobacteria when it is added to the source water before it enters a drinking water treatment plant.

#### 2.2.2. Effect of Coagulation on Cyanobacterial Community Richness and Diversity

To assess the effect of coagulation on the cyanobacterial community richness and diversity, the species richness and Shannon indices were utilized (Figure 3). The richness index is used for estimating the number of different species that are present in an environment, and the Shannon index is used for measuring the homogeneity in the abundance of the different species in a sample [25,26].

In MB, the species richness and Shannon indices did not significantly change in any of the mesocosms after 48 h of their exposure (Appendix A). The average species richness index decreased in the control mesocosms and in the mesocosms containing 20 mgFe/L, while an opposite trend was observed in the mesocosms containing 35 mgFe/L at T48 (Figure 3). This could be attributed to the reduction in the dominant species with a high abundance, such as *Dolichospermum* spp. and *Microcystis* spp., and this could potentially increase the genus richness [14,27]. Furthermore, a prior study demonstrated that a dose of 35 mgFe/L was more efficient than a dose of 20 mgFe/L was in reducing the total cell counts and certain dominant genera such as *Dolichospermum*, *Microcystis*, and *Aphanizomenon* [12]. The Shannon index dropped slightly following the administration of 20 mgFe/L, while it remained almost stable in the control and mesocosms with 35 mgFe/L (Figure 3).

Similarly, there was no significant change in the species richness and the Shannon indices between T0 and T48 in all of the mesocosms in PLSF (Appendix A). The decrease in the species richness index was observed in the coagulated mesocosms, while it increased slightly in the control mesocosms (Figure 3). The decline in the species richness index could be the result of the presence of a genus that could no longer be identified [6]. The Shannon index at T0 (approx. 4) showed that there was a slight change in the diversity profile at T48 (around 4.2) in the control and the 20 mgFe/L mesocosms. In contrast, the Shannon index decreased in the mesocosms with 35 mgFe/L when it was compared with that of the control.

### 2.3. Impact of Environmental Conditions on Cyanobacterial Community Composition in the Mesocosms before and after Coagulation

To assess the impact of the environmental conditions on the cyanobacterial communities in MB and PLSF. A redundancy analysis (RDA) was used to evaluate the relationship between the environmental parameters (Appendix A) and the cyanobacterial communities. The total nitrogen (TN), dissolved nitrogen (DN), dissolved phosphorus (DP), and dissolved organic carbon (DOC) exerted significant effects (ρ < 0.05) on the community profiles in MB (Figure 4).

In MB, in 2018 and 2019, PC1 and PC2 represented 89.4% and 6.12% of the variation, respectively (Figure 4). A relationship between the environmental conditions and the cyanobacterial community composition was not observed in any of the mesocosms from 8–10 August 2018. The result corresponds with that of a previous study which reported that these nutrients (TN, DN, TP, and DOC) did not affect the cyanobacterial community in the supernatant water after coagulation [8]. In contrast, TN, DN, DOC, and DP had a strong impact on *Synechococcus* in the control mesocosms, while DN contributed to the presence of *Microcystis* in the coagulated mesocosms at T48 on 15 August 2019 in MB.

In PLSF, PC1 and PC2 represent 54% and 41.6% of the variation, respectively (Appendix A). A clear association was observed between the DN and *Microcystis* in the control mesocosms on 24–26 July 2019, in the mesocosms with 20 and 35 mgFe/L at T0 on 24 July 2019, and in the mesocosms with 35 mgFe/L on 7 August 2019, where *Microcystis* was the dominant genus. In contrast, the DN was negatively related to *Dolichospermum* at both of the sites. Therefore, the persistence of *Dolichospermum* in the mesocosms could be the result of its environmental resistance [8]. Our result is in accordance with those of previous studies, reporting that the DN may lead to the dominance of *Microcystis* [28,29,30].

## 3. Conclusions

The shotgun metagenomic sequencing method is a more robust method that is used to identify cyanobacterial communities when it is compared with the microscopic methods.In the beginning of the sampling period, *Dolichospermum* was the predomiant genus in Missisquoi Bay, while *Microcystis* was a common genus in all of the mesocosms in Petit Lac St. François.The change in the cyanobacterial composition at the genus level in the mesocosms after two days varied across the studied sites and over the sampling time. Therefore, the choice to use Fe_2_(SO_4_)_3_ as an onsite source-control treatment should be made while considering its impact on the cyanobacterial community structure.*Synechococcus* may be satisfactorily removed in coagulated mesocosms.The intracellular microcystin concentrations were strongly associated with the presence of *Microcystis*.The cyanobacterial community richness and diversity did not change significantly after the coagulation by Fe_2_(SO_4_)_3_ in any of the mesocosms at either of the sites.The dissolved nitrogen content was related to *Microcystis* in both Missisquoi Bay and Petit Lac St. François, while *Synechococcus* was influenced by the total nitrogen, dissolved nitrogen, dissolved organic carbon, and dissolved phosphorus contents.The source water coagulation process is a potential option for removing toxin-producing cyanobacterial species and intracellular toxins from the water column.

## 4. Materials and Methods

### 4.1. Study Sites Description

Missisquoi Bay (MB) and Petit Lac St. François (PLSF) were selected as study sites because of the frequent presence of cyanobacterial blooms [30,31]. MB is located in southern Quebec (Canada; 45°02′23″ N; 73°04′41″ W). It is shallow and eutrophic, with a surface area of 77.5 km^2^, an average depth of 2.8 m, and its groundwater and runoff sources are from predominantly agricultural lands via its tributary rivers [32,33]. MB’s tributaries have high turbidity and excessive concentrations of nitrogen and phosphorus [32,34]. PLSF is located in south-western Quebec (Canada; 45°32′16″ N; 72°02′11″ W; max depth (Z) = 1.8 m). This lake is relatively small, shallow, with half of its watershed being occupied by agricultural land use, and there being high concentrations of phosphorus, resulting in frequent cyanobacterial blooms [30].

### 4.2. Description of the Mesocosm Experiments

The mesocosms were designed according to Wood’s work in 2012 [35]. A mixture of raw lake water and cyanobacterial bloom scum was used to ensure that there were homogeneous initial conditions within the mesocosms. Sets of duplicate mesocosms (*n* = 6) were then filled with that mixed water of 20 L and were left unamended to serve as a control, or amended with two doses of the ferric sulfate coagulant: 20 mgFe/L and 35 mgFe/L. The samples were mixed immediately after the addition of the coagulant [12]. Seven mesocosm experiments were initiated with the appearance of cyanobacterial blooms (Appendix A).

### 4.3. Preparation and Application of Chemical Coagulant

Standard liquid grade Fe_2_(SO_4_)_3_ with 12.2% Fe content was purchased from Kemira Water Solution, Inc. (Varennes, QC, Canada). Two different doses of Fe_2_(SO_4_)_3_ coagulant (20 mgFe/L and 35 mgFe/L) were selected based on a previous study [12].

### 4.4. Sampling and Filtration Procedure

At the beginning (T0) (before adding Fe_2_(SO_4_)_3_) and after two days (T48) of the sampling period, the water in the mesocosms was collected in 1 L sterilized high-density polyethylene (HPDE) bottles. For phytoplankton enumeration, the samples were taken in 40 mL glass vial and preserved in Lugol’s iodine solution. At the sampling site, a 50 mL sample was immediately filtered through a 0.22 μm filter (Millipore Sigma, Oakville, ON, Canada) in sterile conditions that were created with a burner stand. The filters were immediately stored with dry ice during their transfer to the lab where they were frozen at −80 °C for a metagenomic analysis. Samples of 120 mL were filtered on hydrophilic polypropylene; 0.45 µm filters (PALL, Mississauga, ON) were used for the toxins. The samples for the total organic carbon (TOC), total nitrogen (TN), and total phosphorus (TP) were aliquoted without being filtrated. Dissolved organic carbon (DOC) samples were filtered using a 0.45 µm filter membrane (PALL, Mississauga, ON, Canada). A 0.45 μm filter membrane (Millipore Sigma, Oakville, ON, Canada) was used for subsamples of dissolved nitrogen (DN), dissolved phosphorus (DP), and soluble reactive phosphorus (SRP). Genomic samples were taken in triplicate while nutrient samples were taken in duplicate. More detail are available in [12].

Samples for taxonomic cell counts were stored at room temperature in the dark. TN/TP/TOC/DOC samples were stored at 4 °C. Toxin and DN/DP/SRP samples were kept at −25 °C until the analysis was conducted.

### 4.5. Sample Analysis

#### 4.5.1. Taxonomic Cell Counts

Taxonomic cell counts were performanced by using an inverted microscope with a Sedgwick-Rafter counting chamber at 40X magnification (Leica Microsystems GmbH, Wetzlar, Germany) according to [13,36,37].

#### 4.5.2. Nutrient Analysis

Standard methods 353.2 and 350.1 were used for analyzing nitrogen [38]. TOC and DOC were measured based on method 415.1 [39]. Standard method numbers 365.3 and 365.1 [40] were used for phosphorus and phosphate analysis.

Phycocyanin, chlorophyll-*a*, pH, dissolved oxygen (DO), and temperature were measured using a YSI EXO2 water quality multi-parameter sonde (YSI Inc., Yellow Springs, OH, USA) [3,13].

#### 4.5.3. Toxin Analysis

The total microcystins samples were analyzed via a Lemieux oxidation step and an online solid phase extraction (SPE) and salting which were coupled with ultrahigh performance liquid chromatography tandem mass spectrometry (UHPLC-MS/MS). The accuracy was in the range of 93–100%, and it was deemed satisfactory if it was in this range, and the limit of detection was 0.5 ng/L. More details can be found in [12].

#### 4.5.4. DNA Extraction, Metagenomics Preparation and Bioinformatics Analysis

The sample filters were kept in 5 mL Falcon^®^ tubes (VWR, Radnor, PA, USA) at −80 °C and were used for DNA extraction. Total nucleic acid was extracted using this filter by RNeasy^®^ PowerWater^®^ Kit (Qiagen Group, Germantown, MD, USA). Two hundred μL of nuclease-free water and 5 μL of TATAA Universal DNA spike II (TATAA Biocenter AB, Gothenburg, Sweden) were filtered to quantify extraction yields. Dithiothreitol (DTT) was injected into RNeasy^®^ PowerWater^®^ Kit solution PM1 buffer to prevent the formation of disulfide bonds protein residuals. Total nucleic acid was eluted with 60 μL nuclease-free water, and 30 μL was used for the DNA extraction. After resuspending it in 60 μL of nuclease-free water, each DNA sample was quantified using a Qubit^®^ v.2.0 fluorometer (Life Technologies, Burlington, ON, Canada). To perform the pyrosequencing (Roche 454 FLX instrumentation with Titanium chemistry), 30 µL of DNA was sent to the Génome Québec Innovation Centre for sequencing.

An Illumina NovaSeq 6000 S4 (Illumina Inc., San Diego, CA, USA) was used for sequencing DNA libraries. A homemade bioinformatic pipeline was applied for the analysis of paired-end raw reads of 150 base pairs (bp). The raw reads were trimmed to ensure their quality, which was firstly performed using SolexaQA v3.1.7.1 using the default parameters [41]. Next, the trimmed reads that were shorter than 75 nt were analysed. Artificial duplicates were removed using an in-house script based on the screening of identical reads of 20 bp. FragGeneScan-Plus v3.0 was used to predict gene fragments based on the trimmed high-quality reads [42]. Then, predicted fragments of protein were clustered at 90% similarity using cd-hit v4.8.1 [43]. More details are available in [6].

#### 4.5.5. Statistical Analysis

Statistical analysis was performed using R statistical software (R Core Team, 2020). Taxanomic data were normalized by the centred log-ratio transformation using easyCODA (v.0.34.3) [44]. The alpha diversity metrics (Richness and Shannon) were analyzed using vegan package (v.2.6-2) [45], and the similarly matrices were calculated based on the Euclidean distance. The differences between the groups in alpha diversity were determined using Kruskal–Wallis test. Redundancy analysis (RDA) was performed to evaluate the impact of environmental conditions on cyanobacterial community in MB and PLSF, and this was conducted with the vegan package (https://cran.r-project.org/package=vegan, assessed on 10 June 2022).

## Figures and Tables

**Figure 1 toxins-14-00688-f001:**
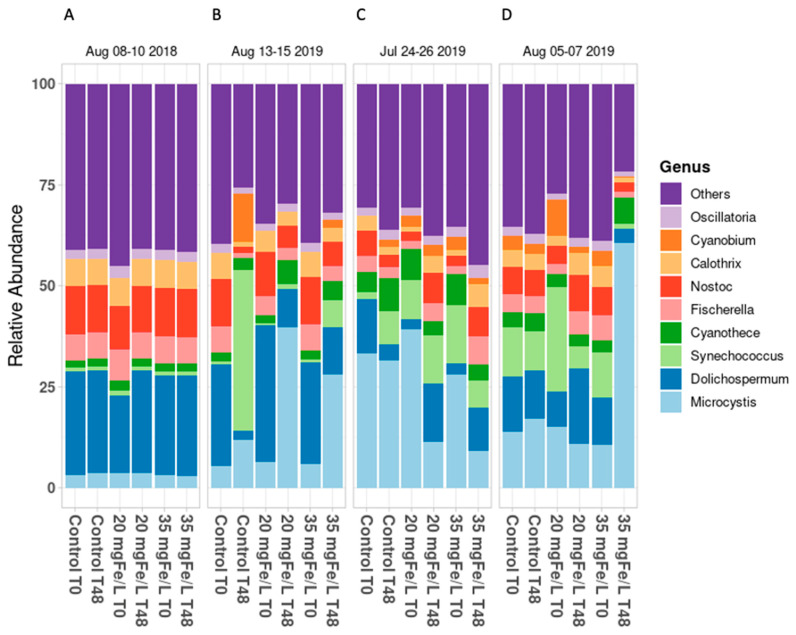
Average cyanobacterial relative abundance at genus level in Missisquoi Bay (**A**,**B**) and Petit Lac St. François (**C**,**D**).

**Figure 2 toxins-14-00688-f002:**
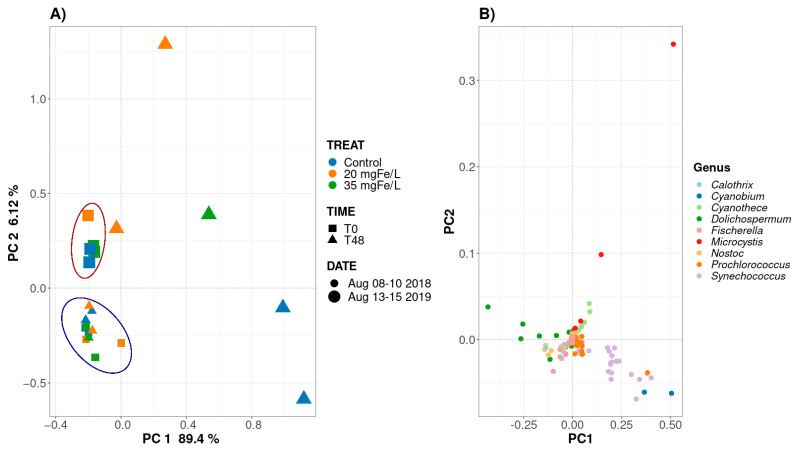
Principal components analysis (PCA) of the normalized relative abundance of cyanobacteria community composition in control, 20 mgFe/L, and 35 mgFe/L mesocosms with respect to genus abundance in Missisquoi Bay. (**A**) PCA analysis of cyanobacterial community following coagulation; (**B**) data plotted following the genus-level classification.

**Figure 3 toxins-14-00688-f003:**
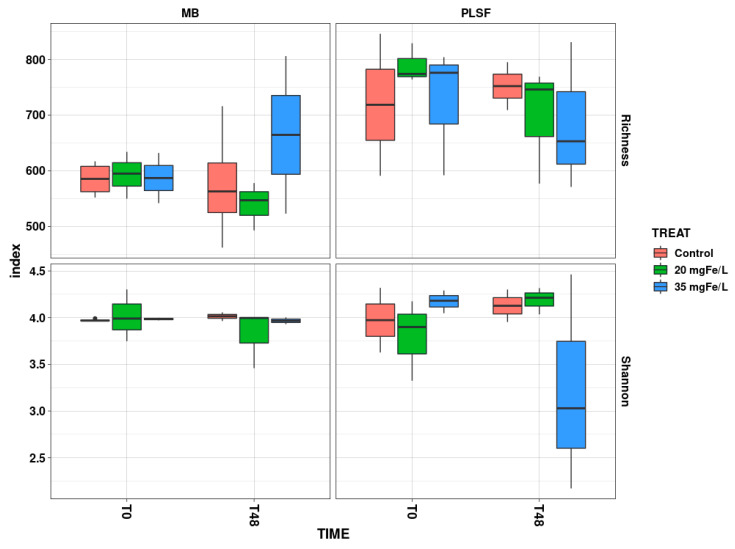
Evaluation of cyanobacterial richness and diversity in control, 20 mgFe/L, and 35 mgFe/L mesocosms in Missisquoi Bay and Petit Lac St. François after two days of treatment. The bottom and top of the box shows the lower and upper quartiles, the band in between them shows the median, and the whiskers show the minimum and maxinum.

**Figure 4 toxins-14-00688-f004:**
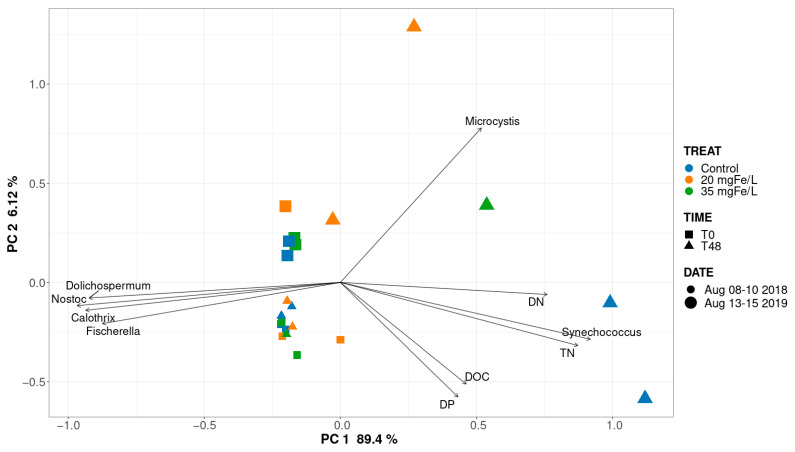
Redundancy analysis (RDA) of environmental parameters with respect to cyanobacterial communities in the control, 20 mgFe/L, and 35 mgFe/L mesocosms in Missisquoi Bay. Only significant parameters (*p* < 0.05) are shown.

## Data Availability

Data is contained within the articles and Appendix A.

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
