# Peer review of "Shotgun Metagenomic Sequencing to Assess Cyanobacterial Community Composition following Coagulation of Cyanobacterial Blooms"

_toxins, 2022, doi:10.3390/toxins14100688_

Round 1

Reviewer 2 Report

As written in the paper, cyanobacteria are an important microbial group having for instance an ability to produce toxins (but they also produce oxygen which we need). Cyanobacteria are a wide group so there are often some species that can grow if the other cyanobacteria have been reduced. The toxicity of the cyanobacterial toxins varies highly and therefore it is important to understand which factors control common cyanobacteria. Thus this work is within the scope of this paper and the paper gives significant new information.

Ferric sulfate (Fe2(SO4)3 is one of the coagulants used in drinking water production mainly for reducing concentrations of organic matter and total phosphorus. After the use of this coagulant, it is interesting to know its effect on cyanobacterial communities in water.

This paper is mainly well written.

In Fig 2 the font for explanation texts is too small. Increase those, especially for time and dates! Change the square to the triangle as you have in the other figures since the small square and the small circle are too similar, but the circle and the triangle are different. You could inform in caption texts the number of parallel tests.

In  Fig 4 use a much darker font for the names of cyanobacteria!   

In references, the font of references 16 and 32 must not be capitalized. Add the bibliographic data to these, since the international readers may not be familiar to find Canadian-USA agreements.  

Scientific names must be italics in references 17, 18, 21, and 27 (at least, check if there are others, too)!    

Reviewer 3 Report

The manuscript is well written and contains new and publishable data. however, some modifications are necessary in order to improve the quality of the manuscript before publication:

Major comments:

- In the results and discussion section, the authors reported that a relationshop between total intracellular microcystin (ΣMCs) and intracellular toxin MC-LR with the genus Microcystis detected through metagenomics was found at T0 and T48 in all mesocosms. However, no data in the materials and methods section about sample preparation and analysis method for these toxins detection?

- It was also interesting that the authors discussed the different results in relation to the nature of species of cyanobacteria, filamentous or colonial, nitrogen-fixing and non-fixing nitrogen.

Minor comments:

- Line 35, the reference [4] is a old reference (2013) and it should be better to add a more recent one concerning the diversity of microcystins, like Bouaïcha et al., 2019 (Toxins 2019, 11, 714; doi:10.3390/toxins11120714).

- Line 48: Similarly, the authors of [10] studied. The sentence should be revised: Similarly, Pei et al. 2017 [10] studied......

- Line 67: With the exception of the study of [12], the sentence should be also revised: With the exception of the study of Le et al. 2021 [12]........

- Line 305: The mesocosms were designed according to the authors of [35]. Should be: The mesocosms were designed according to Wood, 2012 [35].
